# Gingival Margins’ Modifications during Orthodontic Treatment with Invisalign First^®^: A Preliminary Study

**DOI:** 10.3390/children9101423

**Published:** 2022-09-20

**Authors:** Roberta Lione, Francesca Gazzani, Sofia Moretti, Carlotta Danesi, Elisabetta Cretella Lombardo, Chiara Pavoni

**Affiliations:** 1Department of Systems Medicine, University of Rome ‘Tor Vergata’, 00133 Rome, Italy; 2Department of Medicine and Surgery, UniCamillus International Medical University, 00133 Rome, Italy; 3Department of Dentistry, UNSBC, 1000 Tirana, Albania; 4Private Practitioner, 20100 Milan, Italy

**Keywords:** clear aligner treatment, mucogingival modifications, aesthetic, growing patients, arch form development

## Abstract

The aim of the study was to assess modifications of gingival morphology at the end of Phase I treatment with Invisalign First. Eighteen subjects (ten females, eight males, mean age nine years) treated with Invisalign First were selected. The following parameters were measured on intraoral photographs before treatment (T0) and after the first set of aligners (T1) at level of permanent incisors, deciduous canines and molars: gingival margin height (GMH) and deciduous canine inclination (DCI). A paired t-test was used to evaluate T1-T0 changes. The level of significance was established at 5%. Deciduous canines’ GMH showed a major reduction between T1 and T0 accounting for 0.87 mm and 0.86 mm, respectively on the right and left sides. DCI increased for all patients in the interval T0–T1 on both sides, +7.11° on the right and +7.5° on the left. Tooth movement during Invisalign First induced significant modifications of gingival contour resulting in a more harmonious smile.

## 1. Introduction

Invisalign System^®^ was introduced in 1997 as new esthetic and comfortable treatment option to conventional fixed appliances. At first, clear aligner treatment (CAT) was mainly dedicated to adult patients [1]. More recently, the introduction of the Invisalign First System^®^ extended the use of clear aligners to patients between the ages of six and ten years for Phase I treatment [1,2,3]. A correct management of orthodontic Phase I determines early improvements of malocclusions and prepares the patients for Phase II [4]. Clear aligners represent a valid alternative for interceptive treatment in mixed dentition with the addition of several advantages [2]. They provide a better management for both clinicians and patients when compared to traditional appliances: no food limitations, less risk of appliance damage for weekly change, no emergency appointments, simplified oral hygiene, higher patient comfort, easier compliance, and obviously minimal social impact [1,2,3,4,5,6]. From a clinical point of view, one of the main advantages is to enable several movements to occur simultaneously, such as correction of crowding, rotations and reduction of protrusion of teeth during expansion movements [1,7,8]. Moreover, clear aligner treatment minimizes negative effects of orthodontic treatment on dental and periodontal health allowing correct oral hygiene procedures and easier management [9,10]. One of main goals of interceptive treatment during growing phase is to set up an optimal environment for arch form development and changes [11,12,13,14,15,16]. Arch form development obtained during orthodontic treatment with Invisalign First induces significant modifications of mucogingival complex. Clinically, gingival morphology and contour play an important role since the esthetics of soft tissues has always been an important component of a beautiful smile [17,18]. Gingival changes are mainly associated not only with the modifications of teeth bucco-lingual inclinations but also with the derotation and extrusive movements obtained during arch expansion, alignment and leveling of the anterior region. In literature, several studies have assessed gingival margin modifications in permanent dentition before and after fixed appliances revealing that some changes of gingival thickness and keratinized gingival width occurred as a consequence of a better teeth position that allows a different gingival architecture [19,20]. However, to our best knowledge no studies evaluated gingival esthetics in mixed dentition after a Phase 1 interceptive treatment performed only by means of CAT. Hence, the aim of the present prospective investigation was to evaluate the gingival morphology modifications and gingival contour at the end of Phase 1 treatment with Invisalign First System^®^.

## 2. Materials and Methods

The study project was approved by the Ethical Committee at the University of Rome “Tor Vergata” (protocol number 163.20). After declaration of the nature, purpose, and material risks of the procedures, informed consent was obtained from patients’ parents. 18 subjects (10 females and 8 males; mean age 9.4 years ± 1.2) were selected at the Department of Orthodontics, University of Rome “Tor Vergata” according to the following inclusion criteria: European ancestry, dento-alveolar transverse discrepancy of 3–6 mm, early mixed dentition with the presence of permanent incisors, first permanent molars and deciduous canines and molars, mild/moderate crowding, no gingivitis and good compliance during clear aligners treatment. All subjects showed a mesial step or a flush terminal plane molar relationship and a normo-divergence on the vertical plane (SN^GoGn angle from 27° to 37°). The following exclusion criteria were considered: presence of caries, tooth agenesis, supernumerary teeth, cleft lip and/or palate and presence of oral habits.

### 2.1. Treatment Protocol

Treatment protocol for all patients consisted of non-extraction strategies and the absence of any auxiliaries than attachments [21,22]. A standardized expansion protocol with sequential staging was followed for each patient. Treatment protocol also included the initial distorotation of upper first molars followed by sequential expansion of the upper arch and the correction of anterior crowding [9,21]. Additional buccal root torque of 5° was required for latero-posterior segment of the upper arch during expansion movements. Arch expansion amount was of 0.15 mm per stage. Sequential expansion of lower arch was also prescribed to obtain a transversal inter-arches coordination. Alignment and leveling in the anterior region were planned for all subjects according to posterior occlusal plane. All patients were instructed to wear the aligners full time except for meals and hygiene procedures. Aligners were changed every seven days. Clinical checks were planned every four stages to control aligner fitting and attachments position.

### 2.2. Measurements

Digital models in STL files obtained by Itero scanner, intraoral and extraoral photographs were collected before treatment (T0) and at the end of first set of aligners (T1) for each patient. The recorded photographs at two observation times were taken by one operator using the same camera with standardized setting. The interval between the initial and final scans was 7.8 ± 2.4 months. To evaluate the gingival modifications in the T0–T1 interval, the following parameters were measured on intraoral photographs at level of permanent incisors, deciduous canines and molars by using a digital millimeter grid (Figure 1):Gingival margin height (GMH), perpendicular distance between mucogingival line and apical point of the gingival margin;Deciduous Canine Inclination (DCI), angle between long axis of upper deciduous canine and mucogingival line. Mucogingival line was detected by considering mucogingival line of anterior arch segment parallel to occlusal plane.

The additional measurement crown length (CL), referred as distance between the most apical point of the crown and the incisal/occlusal line, was also calculated on digital models (Figure 2) with a 3-dimensional caliber (CaDent Orthocad software). 

Since GMH measured on photographs did not represent the actual size, a multiplication factor was established for calculation of actual variables. The enlargement was obtained by comparing CL of each tooth on the photo with the dimensions of the same tooth on digital model according to the following equation [23]:***Actual GMH*** = GMH (photograph) × CL (digital model)CL (photograph)(1)


### 2.3. Statistical Analysis

A sample size was calculated according to the method proposed by Whitehead et al. [24]. For a standardized effect size of 1 (a clinically relevant change of 0.35 mm with a combined SD of 1.10 mm) for the primary variable GMH, a sample size of 18 subjects was required for an error rate of 5% and a power of 80%. To evaluate intraexaminer reliability, the sample was remeasured 2 weeks after the first evaluation. The reliability of the measurements was assessed by interclass correlation coefficient (ICC). Shapiro–Wilk test was used to test the sample normality. A paired t-test was used to compare the T0–T1 changes with normally distributed data. Level of significance was set at 5% (SPSS, Statistical Package for the Social Sciences, version 18.0 IBM Corp, Chicago, IL, USA). The method error was assessed by randomly selecting 10 subjects. Measurements were repeated on dental casts and photographs within one week by the same operator. The intra-observer reproducibility was analyzed with the intraclass correlation coefficient (ICC). 

## 3. Results

Actual GMH and DCI measurements with mean values, standard deviation and 95% confidence interval are reported in Table 1.

On both sides, a decrement of actual GMH was observed at T1 for each tooth (Figure 3). 

Specifically, deciduous canines showed a major reduction between T1 and T0 accounting for 0.87 mm and 0.86 mm, respectively, on the right and the left sides. Consecutively, first deciduous molars reduced by 0.42 mm on the right side and 0.68 mm on the left, central incisors reduced by 0.41 mm on the right and 0.44 mm on the left whereas lateral incisors reduced by 0.39 mm on the right and 0.46 mm on the left. Second deciduous molars showed a minor reduction, not statistically significant, of 0.22 mm on the right and 0.18 mm on the left. DCI increased (Figure 4) for all patients in the interval T0–T1 on both right side (+7.11°) and (+7.5°). 

As for the measurement errors a variation ranging from 0.1° to 0.3° was observed for the angular measurements and from 0.2 to 0.4 mm for the linear measurements.

## 4. Discussion

The present study assessed the gingival margins modifications after orthodontic treatment with Invisalign First System^®^. Nowadays, social impact is essential for patients of all ages even for children. New generations are very conscious of their appearance. Comfort and aesthetics are the principal requests of young patients and their parents. Proper alignment of teeth induces healthy periodontal tissues with better gingival margin position and better smile design not only in permanent dentition but also in mixed stages [25]. The present study is the first to assess modifications of GMH and DCI in growing patients treated with clear aligners in order to evaluate the esthetic modifications obtained. A limitation involved in the assessment of gingival margin height is that intraoral photographs did not represent the real size of this variable. Therefore, a lot of studies [20,23,26] evaluated gingival modifications including the measurements of clinical crown length (CL) on dental casts. Trentini et al. [27] demonstrated the reliability of photographs and dentals casts for accurate measurements of keratinized tissue width. In the present study, the measurements of GMH were obtained using the equation used in the present investigation. The multiplication factor was used to overcome the problems due to lack of a complete correspondence between intraoral photographs and dental casts. This method of calibration decreased the risks of error.

The magnification correction for the GMH analysis was achieved by comparing it with CL of each tooth on the photo and with the dimensions of the same tooth on the digital model [23]. Some authors have confirmed that individual behavioral factors such as oral hygiene and gingival biotype may modify gingival architecture [23,24,27].

Considering the prospective nature of the present study, great attention was paid to maintain an adequate oral hygiene level. For this reason, oral hygiene instructions were given at any appointment avoiding gingival inflammation [19]. Removable orthodontic appliances, like aligners, enhanced oral hygiene resulting in a lack of plaque accumulation [20]. Therefore, the benefits of orthodontic clear aligner treatment in early mixed dentition must include the favorable impact on the gingival health [27]. Previous studies have analyzed gingival modifications after orthodontic treatment on patients of different ages. Coatom et al. [26] measured the keratinized gingiva width in adolescent patients concluding that before orthodontic therapy it is between 0 to 8.0 mm whereas at the end it is between 0 and 7.7 mm.

Before treatment, upper and lower lateral incisors had the greatest width because they were often displaced to the palatal and lingual side due to crowding. After alignment orthodontic treatment a great decrease in gingival width is observed. Wyrebek et al. [28] stated that in untreated children the width of attached gingiva increased during permanent teeth eruption. This phenomenon could be explained by the fact that the mucogingival junction remained stable and the erupting teeth “pull” the surrounding tissues [28]. The present study showed that there was an improvement of GMH following phase I orthodontic treatment. This change was mostly related to the tooth position in the anterior dental arch. The greatest reduction of GMH was observed at the level of both deciduous canines. According to other authors [23,26] the canines were brought into proper alignment during maxillary expansion in a more vestibular position. The proportion observed between the canine inclination and the remodeling of gingival margins was about 83% with a reduction in GMH of 0.12 mm for each degree of vestibular inclination, mainly determined by expansion protocol. At the level of upper incisors, a mean reduction of 0.43 mm was detected because these teeth underwent a slight vestibular inclination to correct anterior crowding. The limitations of the present study are its preliminary nature, the absence of a control group, and the relatively small sample size. Further investigations are needed to deepen the results obtained and to compare the treated subjects with a control group underwent Phase 1 treatment with traditional appliances. CAT allows several advantages in children in terms of oral hygiene, better aesthetic, more comfort for food and beverage consumption. On the other hand, the use of clear aligners needs a full-time wear to be effective and efficient for malocclusion resolution. The main limitation of CAT in growing patients is represented by the necessity of strict compliance since orthodontic correction is entirely related to the patient’s collaboration.

## 5. Conclusions

Sequential expansion protocol with Invisalign First including distorotation of upper first molars, sequential expansion of the arches, and correction of anterior crowding induced significant modifications of gingival contour resulting in a more harmonious smile. Specifically, these modifications are represented by reduced gingival margin height of upper permanent incisors, upper deciduous canine and molars and increased upper deciduous canine inclination. Attention should be paid to changes in the gingival margin height and in the canine inclination because of the aesthetic effects on smile of growing patients after Invisalign First treatment.

## Figures and Tables

**Figure 1 children-09-01423-f001:**
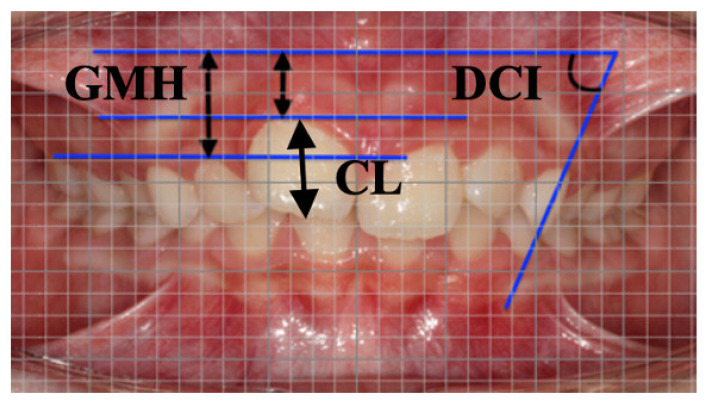
Measurements on intraoral photograph by means of digital millimeter grid at T0. Linear measurements GMH, gingival margin height and CL crown length. Angular measurement DCI, deciduous canine inclination.

**Figure 2 children-09-01423-f002:**
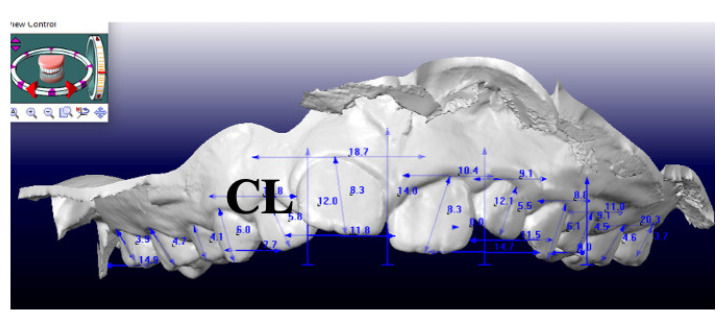
Measurement of CL, crown length on digital models at T0.

**Figure 3 children-09-01423-f003:**
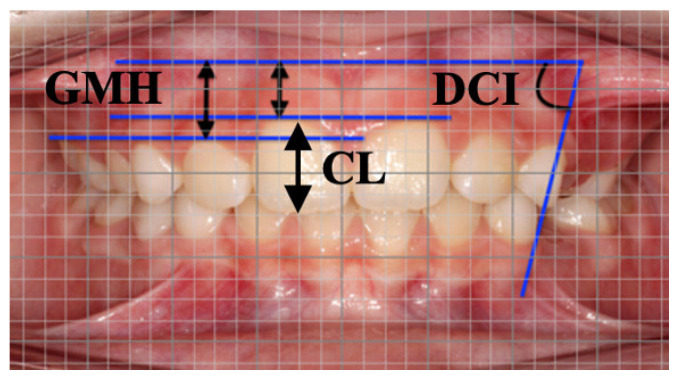
Measurements on intraoral photograph by means of digital millimeter grid at T1. Linear measurements GMH, gingival margin height and CL crown length. Angular measurement DCI, deciduous canine inclination.

**Figure 4 children-09-01423-f004:**
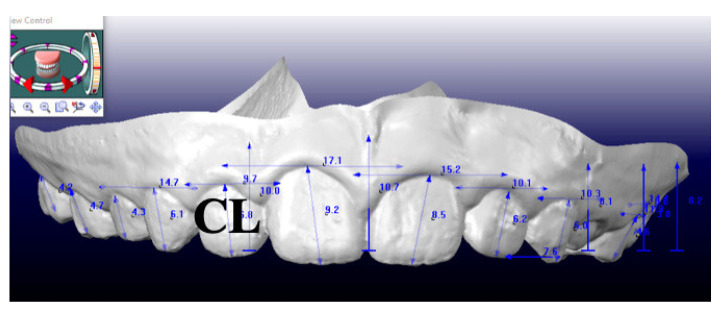
Measurement of CL, crown length on digital models at T1.

**Table 1 children-09-01423-t001:** Descriptive statistics and statistical comparisons of T1-T0 changes (paired *t*-test).

	T0 (*n* = 18, 8 M 10 F)		T1 (*n* = 18, 8 M 10 F)					
Variables	Mean	SD	Mean	SD	Diff	SD	95% CI	*p* Value
**Actual GMH (mm)**								
2° right deciduous molar	4.55	0.59	4.32	0.90	−0.22	0.92	−0.285 to 0.745	NS
1° right deciduous molar	4.55	0.49	4.08	0.93	−0.42	1.04	−0.033 to 0.973	**
Deciduous right canine	5.36	0.59	4.49	0.60	−0.87	0.69	0.466 to 1.273	***
Lateral right incisor	5.18	0.70	4.79	0.74	−0.39	0.99	−0.097 to 0.877	**
Central right incisor	5.21	0.86	4.79	0.67	−0.41	1.07	−0.102 to 0.942	**
Central left incisor	5.66	0.90	5.03	0.85	−0.44	1.20	0.037 to 1.223	**
Lateral left incisor	5.55	0.59	5.09	0.74	−0.46	0.96	0.006 to 0.913	**
Deciduous left canine	5.71	0.62	4.83	0.70	−0.86	0.89	0.432 to 1.327	***
1° left deciduous molar	4.97	0.58	4.52	0.72	−0.68	0.86	0.007 to 0.892	**
2° left deciduous molar	5.25	0.58	5.07	0.97	−0.18	1.11	−0.361 to 0.721	NS
**Canine Inclination (°)**								
Deciduous right canine	83	4.05	89.77	6.61	7.11	6.69	3.056 to 10.483	***
Deciduous left canine	86.27	5.60	93.83	6.24	7.5	5.46	3.543 to 11.576	***

NS Not Significant, 95% CI 95% confidence interval, SD standard deviation. * *p* < 0.05, ** *p* < 0.01, *** *p* < 0.001.

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
