# Peer review of "Gingival Margins’ Modifications during Orthodontic Treatment with Invisalign First®: A Preliminary Study"

_children, 2022, doi:10.3390/children9101423_

Round 1
Reviewer 1 Report (New Reviewer)
Auhors provided an interesting paper and its topic corresponds with current trends in orthodontic treatment- CAT. The paper is well organised. and follows a standard structure (introduction, material and methods, results, discussion and conclusions). I appreciate the figures presenting clearly the measurements. The results are described clearly and completely and correspond with the research method.
I suggest publication after the authors have considered the following minor remarks:
-there is no information about anterio-posterior and vertical problems in inclusion/exclusion criteria
-the authors should also present possible disadvantages and limitations of CAT in children
Author Response
Replies to Reviewers’ comments (ALL CHANGES IN THE TEXT AND TABLES WERE HIGHLIGHTED IN RED FONT).
Reviewer 1
Authors provided an interesting paper and its topic corresponds with current trends in orthodontic treatment- CAT. The paper is well organized and follows a standard structure (introduction, material and methods, results, discussion and conclusions). I appreciate the figures presenting clearly the measurements. The results are described clearly and completely and correspond with the research method.
I suggest publication after the authors have considered the following minor remarks:
-there is no information about anterio-posterior and vertical problems in inclusion/exclusion criteria
Thank you for the comment. The following sentence has been added in Material and Methods (Line 75) section: “All subjects showed a mesial step or a flush terminal plane molar relationship and a normo-divergence on the vertical plane (SN^GoGn angle from 27° to 37°)”.
-the authors should also present possible disadvantages and limitations of CAT in children
We would like to thank the author for the comment. The following section has been added at the end of the Discussion section (Line 204): “CAT allows several advantages in children in terms of oral hygiene, better aesthetic, more comfort for food and beverage consumption. On the other hand, the use of clear aligners needs a full-time wear to be effective and efficient for malocclusion resolution. The main limitation of CAT in growing patients is represented by the necessity of strict compliance since orthodontic correction is entirely related to the patient’s collaboration”
Reviewer 2 Report (New Reviewer)
It's an interesting study concerning a relatively new but quite popular treatment.
I have some concerns about the methodology. In particular, the conditions under which the photographs were taken have not been clearly described. If the photos were not taken under the same conditions the measurements are not comparable. Consequently, the results are not objective.
Moreover, it has not been clarified how many observers performed the measurements and how many times. The intra and inter-observer agreement has not been tested.
A more shophisticated statistical analysis should have been performed.
Author Response
Reviewer 2
It's an interesting study concerning a relatively new but quite popular treatment.
I have some concerns about the methodology. In particular, the conditions under which the photographs were taken have not been clearly described. If the photos were not taken under the same conditions the measurements are not comparable. Consequently, the results are not objective.
We would like to thank you for the comment. The following sentence has been added in the Material and Methods section (line 96): “The recorded photographs at two observation times were taken by one operator using the same camera with standardized setting”.
Moreover, it has been specified in discussion section (line 178) as follow: “The multiplication factor was used to overcome the problems due to lack of a complete correspondence between intraoral photographs and dental casts. This method of calibration using decreased the risks of error.
Moreover, it has not been clarified how many observers performed the measurements and how many times. The intra and inter-observer agreement has not been tested. A more shophisticated statistical analysis should have been performed.
Thanks for the comment. The following sentence has been added at the end of Materials and methods section (Line 135): “The method error was assessed by randomly selecting 10 subjects. Measurements were repeated on dental casts and photographs within 2 one week by the same operator. The intra-observer reproducibility was analyzed with the intraclass correlation coefficient (ICC)”.
In the Results section (Line 164) the following sentence has been added: “As for the measurement errors a variation ranging from 0.1° to 0.3° was observed for the angular measurements and from 0.2 to 0.4 mm for the linear measurements”.
Round 2
Reviewer 2 Report (New Reviewer)
It seems that the authors have considered all the comments
This manuscript is a resubmission of an earlier submission. The following is a list of the peer review reports and author responses from that submission.
Round 1
Reviewer 1 Report
Dear authors
I find your paper very interesting, the argument is very actually and the study is well designed.
I recommend you to add a part in your introduction reguarding menagement of oral hygiene (home and in office) during the Invisalign therapy with a particular focus on the attachments daily cleaning and their remineralization.
I suggest you to add this article in the references "PRE-BONDING PROPHYLAXIS AND BRACKETS DETACHMENT: AN EXPERIMENTAL COMPARISON OF DIFFERENT METHODS Lanteri Valentina DDS, MS, PhD Student, Segù Marzia DDS, MS, PhD, Doldi Jennifer DH, Butera Andrea DH University of Pavia, Italy International Journal of Clinical Dentistry Vol 7 Issue2 2014"
Author Response
Replies to Reviewers’ comments (ALL CHANGES IN THE TEXT AND TABLES WERE HIGHLIGHTED IN RED FONT).
Reviewer 1
- I recommend you to add a part in your introduction regarding management of oral hygiene (home and in office) during the Invisalign therapy with a particular focus on the attachments daily cleaning and their remineralization.
Thanks to the reviewer for the comment. The following sentences and references have been added in the introduction section (line 10): “Moreover, clear aligner treatment can minimize the orthodontics-related negative effects on dental and periodontal health allowing easier oral hygiene procedures and management [8,9]”.
“8. Rossini G, Parrini S, Castroflorio T, Deregibus A, Debernardi CL (2015). Periodontal health during clear aligners treatment: a systematic re- view. Eur J Orthod 37:539-43;
- Chhibber A, Agarwal S, Yadav S, Kuo CL, Upadhyay M (2018). Which orthodontic appliance is best for oral hygiene? A randomized clinical trial. Am J Orthod Dentofacial Orthop. 153 (2): 175-183”
- I suggest you to add this article in the references "PRE-BONDING PROPHYLAXIS AND BRACKETS DETACHMENT: AN EXPERIMENTAL COMPARISON OF DIFFERENT METHODS Lanteri Valentina DDS, MS, PhD Student, Segù Marzia DDS, MS, PhD, Doldi Jennifer DH, Butera Andrea DH University of Pavia, Italy International Journal of Clinical Dentistry Vol 7 Issue2 2014"
We would like to thank the reviewer for the comment. The suggested article has been added in the references.
Reviewer 2
Thank you for the opportunity to review this manuscript. In this paper, the authors evaluated the modifications of gingival morphology and gingival contour at the end of treatment with Invisalign First. Although this manuscript conception is interesting, there are two flaws that make me discourage publication on Children:
- Why did the authors not consider obtaining clinical measurements representing the real size?
Measurements were revealed in photographs that does not represent the actual size of the variables measured. For this reason, a multiplication factor was established.
- Moreover, periodontal indices could have been obtained during the treatment phases (from baseline to final).
We would like to thank for the comment. The main objective of the paper was to evaluate gingival modifications considering tooth movements determined by the treatment. Periodontal indices was not considered.
- As the authors declared it to be a prospective study, why did they not consider obtaining clinical periodontal evaluations since they aimed to evaluate changes in the gingival margin with treatment?
We would like to thank for the comment.
- The authors could have considered including a control group. Were the results obtained because of the type of appliance?
We would like to thank for the comment. Further investigation probably will consider a control group. However, a control group treated with RME could be not comparable since this treatment does not include the immediate alignment of anterior teeth.

Reviewer 2 Report
Thank you for the opportunity to review this manuscript. In this paper, the authors evaluated the modifications of gingival morphology and gingival contour at the end of treatment with Invisalign First. Although this manuscript conception is interesting, there are two flaws that make me discourage publication on Children:
- Why did the authors not consider obtaining clinical measurements representing the real size? Moreover, periodontal indices could have been obtained during the treatment phases (from baseline to final). As the authors declared it to be a prospective study, why did they not consider obtaining clinical periodontal evaluations since they aimed to evaluate changes in the gingival margin with treatment?
- The authors could have considered including a control group. Were the results obtained because of the type of appliance?
Author Response
Replies to Reviewers’ comments (ALL CHANGES IN THE TEXT AND TABLES WERE HIGHLIGHTED IN RED FONT).
Thank you for the opportunity to review this manuscript. In this paper, the authors evaluated the modifications of gingival morphology and gingival contour at the end of treatment with Invisalign First. Although this manuscript conception is interesting, there are two flaws that make me discourage publication on Children:
- Why did the authors not consider obtaining clinical measurements representing the real size?
Measurements were revealed in photographs that does not represent the actual size of the variables measured. For this reason, a multiplication factor was established.
- Moreover, periodontal indices could have been obtained during the treatment phases (from baseline to final).
We would like to thank for the comment. The main objective of the paper was to evaluate gingival modifications considering tooth movements determined by the treatment. Periodontal indices was not considered.
- As the authors declared it to be a prospective study, why did they not consider obtaining clinical periodontal evaluations since they aimed to evaluate changes in the gingival margin with treatment?
We would like to thank for the comment.
- The authors could have considered including a control group. Were the results obtained because of the type of appliance?
We would like to thank for the comment. Further investigation probably will consider a control group. However, a control group treated with RME could be not comparable since this treatment does not include the immediate alignment of anterior teeth.

Reviewer 3 Report
Dear authors,
Abstract
Begin with a capital letter
Statistical significance should not be mentioned
Do not start a sentence with “18”
How did you measure the parameters on photographs?
There is a big bias in doing this.
Introduction
Must be improved
Definition of Invisalign is missing
“Clear aligners represent a valid alternative for interceptive treatment in mixed dentition” is not true. In the mixed dentition functional appliances are used
“One of main goals of interceptive treatment during growing face is represented by transversal expansion and arch form changes” – Invisalign does not do that
This statement is too brave! “However, no studies evaluated gingival esthetics in mixed dentition after a Phase 1 interceptive treatment”
Please define the objectives more clearly.
Materials and methods
Did you calculate NNT? You must calculate the power of the sample and the needed subjects to treat.
I have some doubts about your statistics.
“A sample size was calculated according to the method proposed by Whitehead et al. [24]. For a standardized effect size of 1 (a clinically relevant change of 0.35 mm with a combined SD of 1.10 mm) for the primary outcome variable GMH, a sample size of 18 subjects were required for a type I error rate of 5% and a power of 80%”
Looks like 18 subjects are too few. How did you establish SD = 1.10???
Confidence Level: 95%
Margin of Error:
Population Proportion:
Population Size:
You did not have controls!
P value must be shown, not just *
How did you measure deciduous canine inclination? You must have a stable reference line!
Figure captions should be presented below the picture.
The discussion is poorly written.
Conclusions are not sustained by the results.
References are not in the journal style
Round 2
Reviewer 2 Report
Dear authors, I would consider all the mentioned topics in the review process to stablish new scientific research.
Author Response
We would like to thank for the comments. The manuscript has been checked and modified according to the suggestions of the reviewer.
Round 3
Reviewer 2 Report
Dear authors, I would consider all the mentioned topics in the review process to stablish new scientific research.